# From Childhood Residential Green space to Adult Mental Wellbeing: A Pathway Analysis among Chinese Adults

**DOI:** 10.3390/bs12030084

**Published:** 2022-03-17

**Authors:** Hansen Li, Matthew H. E. M. Browning, Yang Cao, Guodong Zhang

**Affiliations:** 1Institute of Sports Science, College of Physical Education, Southwest University, Chongqing 400715, China; lhs510416413@email.swu.edu.cn; 2Department of Park, Recreation and Tourism Management, Clemson University, Clemson, SC 29634, USA; mhb2@clemson.edu; 3Clinical Epidemiology and Biostatistics, School of Medical Sciences, Örebro University, Orebro 70182, Sweden; 4Unit of Integrative Epidemiology, Institute of Environmental Medicine, Karolinska Institutet, Stockholm 1317177, Sweden

**Keywords:** green exercise, children, residential green space, mental wellbeing, nature connectedness

## Abstract

Residential green spaces, arguably the most accessible type of urban green space, may have lasting impacts on children and even change their lives later in adulthood. However, the potential pathways from childhood residential green space to adulthood mental wellbeing are not well understood. Therefore, we conducted a questionnaire survey among Chinese adults (N = 770) in September 2021 to capture data on subjective measures of residential green space and nature contact during childhood, and nature connectedness, nature contact, and mental wellbeing during adulthood. Structural equation modeling (SEM) was employed to examine theoretical pathways between childhood residential green space and adult mental wellbeing. The results suggest that childhood residential green space positively predicts childhood nature contact and also has direct and indirect positive impacts on nature contact, nature connectedness, and mental wellbeing during adulthood. These findings advance understanding of the long-term impacts of childhood residential green space. Policymakers are advised to prioritize residential greening as well as other recreational facilities for children when planning health-promoting environments in urban spaces. Due to limitations in our study design, we also advise future studies to re-examine and extend the framework documented here.

## 1. Introduction

Contacting with nature is known to benefit mental and physical health. However, owing to global urbanization, more than half of the world’s population lives in cities, and this proportion is projected to reach nearly 70% by 2050 [1]. People today are constantly staying indoors, especially children, and are less active in natural environments than previous generations, bringing up concerns about a growing “disconnection” from nature [2,3].

In this context, urban green spaces are valuable for maintaining connections between people and nature. Urban green spaces can offer less polluted and attractive environments for urban residents to carry out various activities, thus enabling them to interact with natural elements more physically than an entirely “gray” urban environment [4,5]. Among the various types of urban green spaces, residential (near/at home) green spaces may be popular because travel distance is a critical factor that affects green space utilization, and a closer green space usually means more frequent visits [6,7].

So far, the health benefits of residential green space have been widely documented, such as lower risk of depression and higher levels of wellbeing [8,9]. Studies have also explored the pathways underneath these health benefits [10,11,12]. Notably, some recent studies have identified the impact of childhood residential green space on mental health issues in adulthood [13,14], implying a long-term impact of residential green space. These findings collectively enrich the previous knowledge concerning the impacts of childhood experiences and environments on mental health in later life [15,16,17]. Nevertheless, since the long-term impact of green space remains an emerging topic, the potential pathways are not fully understood, and the impact across time requires theoretical clarification.

### 1.1. Role of Residential Green Space on Children’s Nature Contact

Residential environments are known to affect children’s health and behavior [18,19,20,21]. For example, higher levels of residential green space have been associated with lower problematic behaviors, higher intelligence, and better neurodevelopment [21,22,23]. However, these studies have concentrated on the conditions of residential green space, while children’s physical contact with nature has not been fully considered. Theoretically, green space may motivate children to perform activities inside by sheltering them from heat and urban pollution [4,24]. According to a Chinese report, children in Shang, China, mainly perform outdoor activities in residential green space, indicating the role of residential green space in providing children places for nature contact [25]. Based on these clues, we hypothesized a pathway from residential green space to children’s contact with nature.

### 1.2. From Childhood Nature Contact to Adult Mental Wellbeing

Though we have identified limited evidence for a pathway linking childhood nature contact and adult mental wellbeing, we posited two indirect pathways based on existing clues.

The first indirect pathway is mediated by adult nature contact. As described above, childhood experiences may affect adult behaviors. Similarly, childhood experiences in nature may affect behavior that spurs nature contact. For example, Thompson, et al. [26] found that frequent visits to green spaces in childhood were strongly associated with being prepared to visit green spaces alone as an adult. Holt, et al. [27] found that daily interactions with green space in childhood were associated with frequent green space use as a university student. Similar pathways linking childhood and adulthood were confirmed in pathway analyses [28,29]. Hence, we established a pathway between childhood nature contact and adulthood nature contact. Since contact with nature is well-known to benefit mental wellbeing, we further established a pathway between adult nature contact and adult mental wellbeing [30,31].

The second indirect pathway is mediated by nature connectedness. Nature connectedness refers to an emotional attachment to the natural world [32]. According to previous research, nature connectedness can be developed through nature exposure [33,34], and childhood is a critical time window for cultivating nature connectedness [2]. Therefore, children’s nature contact may help them gain a stronger “connectedness” with nature. In addition, as nature connectedness may not readily decay over time [35], children who developed nature connectedness may be able to maintain this emotional attachment when they grow up. For these reasons, we established a pathway between childhood nature contact and adult nature connectedness. According to the biophilia hypothesis, individuals’ psychological health is associated with their relationship to nature [36]. Some previous studies have suggested potential impacts of nature connectedness on wellbeing [37,38]. Accordingly, we established a pathway between nature connectedness and mental wellbeing.

### 1.3. Conceptual Model

Based on the above-hypothesized pathways, we developed our conceptual framework that links residential green space in childhood and mental wellbeing in adulthood (Figure 1). Several demographic variables such as gender, age, and income are also considered in the framework. These variables have been reported to impact mental health [39,40,41], as well as psychological traits, such as nature connectedness [42]. Regarding gender, there is an assumption that women tend to be more sensitive to aspects related to the natural world [43]. Gender is also associated with certain social roles, which may influence their time at home or visiting outdoor environments and, therefore, chances for nature contact [44]. In addition, gender may influence willingness to engage in outdoor physical activities and chances for nature contact [45]. Regarding age, we can expect a similar impact of social roles, life stages, and sensitivity to aspects of the natural world to vary by age. Regarding income, family-level socioeconomic status (SES) may impact the health benefits of exposure to green spaces [46]. Last, regarding urbanization, population density can affect residents’ chances for green space access and associated health benefits [24,47,48]. Based on the framework, we hypothesized the following:(1)Childhood residential green space may affect adult nature contact and nature connectedness through childhood nature contact;(2)Childhood residential green space may affect adult mental wellbeing through childhood nature contact, adult nature contact, and adult nature connectedness.
Figure 1Conceptual framework linking childhood residential green space to adult mental wellbeing.
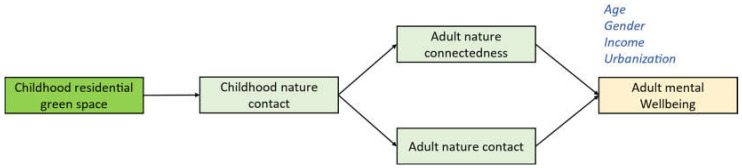


## 2. Materials and Methods

### 2.1. Study Design and Population

We conducted a cross-sectional investigation during September 2021. The target population was Chinese citizens aged 18 years or older without restriction for the size of cities that respondents lived in. The study was approved and supervised by the Ethics Review Board of Southwest University, China.

### 2.2. Procedure

Our research team distributed messages via online social media including WeChat and Tencent QQ to recruit participants. We described the study’s goal as examining the relationship between green space in urban environments and the public health of urban residents. The details of the research questions were not disclosed during the recruitment. We offered a compensation of RMB 4 per participant for completing the online questionnaire to maximize the response rate.

The participants used an identity authentication platform to complete the questionnaire. They were asked to sign a consent form for the investigation and finish the questionnaire within 24 h. In total, 752 successfully submitted questionnaires were collected and included for analysis.

### 2.3. Measurements

#### 2.3.1. Childhood Residential Green Space

Given the difficulties of recalling childhood, we used a single question to investigate the greenness of childhood residential green space, which was used in previous studies to capture perceived green space or nature exposure [10]. We investigated nature exposure during childhood within a predefined timeframe [49], which aligns with the age between 5 and 10 years old. The specific statement was:

“Please indicate the degree of greenness in your residential neighborhood when you were between 5 and 10 years old. Greenness refers to trees or any plants; the residential neighborhood refers to the area around your residences that is within a very few minutes walk”.

We used a 5-level response scale to collect answers, where 1 = extremely low (almost no greenness), 2 = low, 3 = moderate, 4 = high, and 5 = extremely high (the open space is basically full of greenness).

#### 2.3.2. Childhood Nature Contact

We followed the approach by Larson, et al. [50] and Rosa, et al. [28] to investigate the contact with nature during childhood. We defined contact as intentional activities in a natural environment, including physically visiting, playing, and/or engaging in sport. In line with the above time frame, we asked the following question:

“Between 5 and 10 years old, how frequently did you participate in leisure activities in contact with nature? Leisure activities in nature may include visiting natural places, playing soccer or volleyball at the beach, swimming, surfing, camping, hiking, etc.”.

Answers were obtained on a 5-level response scale (1 = never, 2 = a few days, 3 = half of days, 4 = most of days, 5 = almost every day).

#### 2.3.3. Adult Nature Contact

Following the approach for childhood nature contact, a similar question was deployed to investigate respondents’ current contact with nature:

“Currently, how frequently do you participate in leisure activities in contact with nature? Leisure activities in nature may include visiting natural places, playing soccer or volleyball at the beach, swimming, surfing, camping, hiking, etc.”. Answers were obtained on a 5-level response scale (1 = never, 2 = a few days, 3 = half of days, 4 = most of days, 5 = almost every day).

#### 2.3.4. Adult Nature Connectedness

We employed a short version of the Connectedness to Nature Scale (CNS-7) to measure nature connectedness [51]. Participants rated seven items on their affective connections with nature, such as “I often feel a kinship with animals and plants”. Answers were obtained on a 5-point Likert-type scale (1 = strongly disagree, 5 = strongly agree). The validity and reliability of the CNS-7 have been confirmed in several studies [51,52]. The CNS-7 showed good internal consistency in the current study (Cronbach α > 0.9).

#### 2.3.5. Adult Mental Wellbeing

The WHO-5 wellbeing index was used as a reliable and validated scale to measure mental wellbeing [53]. This scale consists of seven items, such as “I have felt calm and relaxed” and “I woke up feeling fresh and rested”. Answers were obtained on a 5-point Likert-type scale (0 = never, 4 = every time). The WHO-5 has been previously employed to investigate mental wellbeing under the impacts of green space exposure [54]. To better associate mental wellbeing with green exercise levels, we used “over the last month” as the time frame for these questions. We used a Chinese version of WHO-5, and it showed good internal consistency in the current study (Cronbach α > 0.9).

#### 2.3.6. Sociodemographic Characteristics

We controlled for three sociodemographic characteristics at the individual-level: age (in years), gender (male = 1, female = 2), and monthly income (1 = no income, 2 = income ≤ RMB 3000, 3 = RMB 3001–5000, 4 = RMB 5001–7000, 5 = RMB 7001–9000, 6 = RMB 9001–11,000, 7 = RMB 11,001–13,000, 8 = RMB 13,001–15,000, 9 = more than RMB 15,000). One RMB is approximately 0.8 USD or 0.7 EUR.

#### 2.3.7. Urbanization Rate

The urbanization rate was obtained from the National Bureau of Statistics of China. It is the ratio of urban populations divided by the total populations of a specific province. For example, an urbanization rate of 0.5 indicates that 50% of the population is settled in urban areas. Our data was calculated based on the national investigation at the end of 2020.

## 3. Analysis

### 3.1. Internal Reliability of Employed Instrument and Bivariate Correlations

Internal reliability levels of the CNS-7 and WHO-5 were analyzed using Cronbach’s alpha. Spearman and point-biserial correlations (for gender) were employed to detect general patterns of associations within the data.

### 3.2. Application of Structural Equations Modeling

Structural equations modeling (SEM) was employed to investigate the hypothesized directional pathways in the presented hypothesized framework. The initial model (M0) was constructed as shown in Figure 2.

According to Bagozzi and Yi [55], the sample size for SEM should be twice the number of model parameters. Our sample size was adequate based on the parameters required to estimate in our framework (*n* = 37). Guided by previous pathway analysis [11,12], nature connectedness and mental wellbeing were processed as continuous summary scores. Therefore, no latent variable was involved in the current study. A Variance Inflation Factor (VIF) value smaller than 5.0 was considered evidence for the absence of multicollinearity. Based on this rule, no multicollinearity was observed for the independent variables (VIF < 3.0) [56].

We used a maximum likelihood estimator with the bootstrap generated (10,000 samples) confidence intervals, standard errors, and significant levels for all pathways [57,58,59]. The goodness of fit was assessed using the following indices: χ^2^/df < 5.00; root mean square error of approximation (RMSEA) < 0.08; goodness-of-fit index (GFI) > 0.90; adjusted goodness of fit index (AGFI) > 0.90; Bentler’s comparative fit index (CFI) > 0.90; and Bentler–Bonett normed fit index (NFI) > 0.90.

### 3.3. Model Adjustment

The initial model (M0) showed a poor fit to the data (χ^2^/df = 14.17, GFI = 0.95, AGFI = 0.83, NFI = 0.81, CFI = 0.82, RMSEA = 0.13). After that, we modified the model by establishing new pathways between the core variables according to modification indices and potential causalities. The modified model (M1) showed an acceptable fit to the data (χ^2^/df = 2.87, NFI = 0.97, GFI = 0.99, AGFI = 0.96, CFI = 0.98, RMSEA = 0.50). Thus, this model was selected as the final model, and non-significant pathways were retained.

### 3.4. Sensitivity Analysis

We established competing models by altering theoretical causalities of the final model and comparing them with the final model using Akaike information criterion (AIC) and Bayesian Information Criterion (BIC) values [60]. In addition, we conducted a multigroup pathway analysis to examine if the final model applied differentially to men and women. Given the generation difference, a multigroup pathway analysis was also conducted for data stratified by the median age (28 years).

## 4. Results

### 4.1. Characteristics of the Participants

The average age of the participants was 31.15 years, and 43.62% were females. More than half of the participants reported a high or extremely high level of greenness in their childhood residential neighborhood. Additionally, most of the participants (28.86%) reported that they contacted nature almost every day during their childhood. However, only 10.11% of the participants reported nature contact almost every day as an adult (Table 1).

### 4.2. Correlations between the Measurements

Table 2 presents the bivariate correlations between all measured variables. Childhood residential green space and nature connectedness were positively correlated with all core variables. Females tended to have higher nature connectedness than males. Regarding control variables, age was positively correlated with childhood nature contact, adult nature contact, and nature connectedness. Income was positively correlated with all core variables. Urbanization rate was positively correlated with adult nature contact.

### 4.3. SEM Model

In the final model, all the hypothesized direct pathways in the conceptual framework (Figure 1) were statistically significant (Figure 3). In addition, the new pathways established during the model adjustment were found to be statistically significant. Specifically, we found direct pathways between childhood residential green space and adult nature contact (β = 0.16, *p* < 0.001), adult nature connectedness (β = 0.13, *p* = 0.001), and adult mental wellbeing (β = 0.09, *p* = 0.001). We also identified a direct pathway between adult nature contact and adult nature connectedness (β = 0.40, *p* < 0.001).

We found that childhood residential green space had significant total impacts on childhood nature contact (β = 0.55, *p* < 0.001), adult nature contact (β = 0.30, *p* < 0.001), adult nature connectedness (β = 0.30, *p* < 0.001), and adult mental wellbeing (β = 0.22, *p* < 0.001) (Table 3). The indirect impacts of childhood residential green space on adult nature contact (β = 0.15, *p* < 0.001), adult nature connectedness (β = 0.16, *p* < 0.001), and adult mental wellbeing (β = 0.12, *p* < 0.001) were also statistically significant (Table 3).

We also examined the specific indirect pathways between childhood residential green space and adult mental wellbeing (Table 4). Although all pathways were statistically significant, the pathway mediated by childhood nature contact and adult nature connectedness was relatively weak (β = 0.008, *p* = 0.033). The pathway mediated by adult nature contact showed the strongest effect (β = 0.029, *p* < 0.001).

### 4.4. Sensitivity Analysis

We compared the final model with two competing models based on the extra hypotheses: (1) adults with higher mental wellbeing are more willing to contact nature (model C1), and (2) higher nature connectedness may encourage more frequent nature contact (model C2) (Appendix A). The competing models showed acceptable fits, but the final model retained the lowest AIC and BIC value, ultimately indicating a better fit (Appendix A).

In models stratified by gender, the baseline model (unconstrained model) and constrained model (structural weights were constrained) showed acceptable fits to the data for both males and females without statistically significant difference between the two models (Δχ^2^ = 22.71, *p* = 0.09). This finding indicated that the final model was suitable for both genders.

A statistical difference was found when stratifying the models by age (Δχ^2^ = 31.73, *p* = 0.01). We compared pathways in the two models and found that the difference resulted from changes in some confounding pathways. The pathways between the core variables remained statistically significant in both groups, indicating that the main framework could fit both age groups well.

## 5. Discussion

We examined the potential causal pathways between childhood residential green space and adult mental wellbeing. We found that childhood residential green space promoted childhood nature contact, which contributed to adult nature contact, nature connectedness, and mental wellbeing.

### 5.1. Role of Childhood Residential Green space on Adult Nature Contact and Nature Connectedness

We found two indirect pathways between childhood residential green space and adult nature contact and nature connectedness. Childhood nature contact was a common mediator in both. These findings support our first hypothesis, indicating a mechanism to understand the long-term impact of childhood residential green space. In previous pathway analysis, children’s experience in nature was found to promote nature connectedness and nature-based recreational activities as an adult [29]. Our findings confirm this pathway and extend it to childhood residential green space, underling the role of childhood residential green space in offering children places for nature contact.

We found several pathways that were not assumed in our hypothesized framework. First, we found direct impacts of childhood residential green space on adult nature contact and nature connectedness. These results are not entirely surprising, however, since residential green space may directly and continuously impact residents by buffering environmental disturbances that can be easily perceived, such as noise and air pollution [11,61]. In addition, residential green space is the type of green space closest to the home. Thereby, some remote exposures, such as viewing green space through windows, may improve mental health [62,63]. These impacts may affect children and change their attitudes towards nature and nature-related activities without physically entering residential green spaces.

Another unexpected finding is the pathway between adult nature contact and adult nature connectedness. Theoretically, a reciprocal relationship may exist between the two variables because nature connectedness is believed to be fostered through nature contact [32]. Meanwhile, nature connectedness may be a motivator for people to engage with nature [29,64]. Based on our cross-sectional design, the time-sequences of the investigated variables, and the results of sensitivity analysis, a pathway from adult nature contact to adult nature connectedness appears more logical. Taken together, our results suggest that childhood and recent nature contact may both contribute to nature connectedness. These findings partially support a previous study, where childhood nature experience promoted adult recreational nature contact and, in turn, enhanced nature connectedness [28]. Our findings also imply a relationship between nature connectedness and lifetime nature-related experiences [65].

### 5.2. Role of Childhood Residential Green Space on Adult Mental Wellbeing

We found that childhood residential green space had indirect positive impacts on adult mental wellbeing. Adult nature connectedness and adult nature contact were critical mediators for the indirect effects. These results support our second hypothesis. The identified indirect pathways may partially explain previous findings that childhood residential green space may predict mental health conditions later in life [13,14]. The effects of childhood experiences encourage health promoting behaviors and psychosocial characteristics across the life course [15,66,67]. For example, Engemann, et al. [13] speculated that stress-processing abilities developed through exposure to childhood green space could explain their long-term health benefits. Based on our findings, nature connectedness and nature contact may be durative health-related factors that connect childhood and adulthood.

In addition to the observed indirect pathways, a direct pathway was found between residential green space and adult mental wellbeing. Although the effect of this direct pathway was subtle (β = 0.09, *p* < 0.05), it implies a long-term impact of residential green space across the life course. This may be explained by the potential for stress reduction through green space exposure [68,69]. As persistent stress in childhood may negatively change the nervous system and make individuals vulnerable to illness [67], residential green space may buffer against this harmful process and protect mental health in later life.

It is noteworthy that our study was conducted during the COVID-19 pandemic. In the past two years, the novel virus has threatened physical and mental health worldwide. In this context, nature contact has been advocated as a way to cope with mental health issues [70]. Although social blockades may limit residents’ access to green space [71], it has also been reported that some residents increased green space utilization for recreation [72]. Some previous non-users of urban green space began using these spaces during the pandemic [73]. The conditions in China are currently understudied. Nevertheless, it is reasonable to assume that some people in China may visit green spaces for mental health benefits because evidence suggests the Chinese demonstrated high demand for green spaces during the pandemic [74], and most green spaces were accessible during the period leading up to the current study. Therefore, the connection between adult nature contact and mental wellbeing may have been strengthened during the pandemic, which may have increased the impact of childhood residential green space on adult mental wellbeing.

### 5.3. Implications and Suggestions

This study underlines the role of residential green space as a source of childhood nature contact and adult nature contact, nature connectedness, and mental wellbeing. There is a rising demand for enabling children to contact nature and build emotional connections with nature [75]. This demand has begun to receive more attention in China, as China has been urbanizing faster than other countries in the past decade [76]. In a previous study, Zhang, et al. [77] recommended that policymakers encourage the public to cultivate emotional engagement with nature and prioritize the improvement of nature connectedness in children. Rosa, et al. [29] concluded that one strategy to promote nature-based recreations is access to safe green spaces near residential areas and workplaces. Based on the information that Chinese children may prefer to perform physical activity in residential green space, the strategy stated by Rosa, et al. [29] seems practical for children in China. Thus, policymakers are recommended to prioritize greening projects in residential areas. To ensure the safety and convenience of children’s activities, specialist recreational facilities in green spaces need more attention.

Moreover, our findings may provide guidance for schools. Children spend more waking hours in school than at home, especially in developing countries such as China. Therefore, the greenness in and around school campuses is essential to provide children with nature contact. The current study focused on children between 5 and 10 years old, which is approximately equivalent to the age range of elementary school. According to a previous study, the years before age 11 may be more critical for fostering nature connectedness, indicating the significance of nature connectedness promotion in elementary school [78]. Dedicated environmental education programming is a popular method to foster nature connectedness in children [78,79,80]. However, it may not be suitable for Chinese children because their schedule is full. Incorporating nature contact into pre-existing school curricula may be more realistic. For example, building a green playground for physical education classes can be helpful. In addition, informing parents of the benefits of nature contact and encouraging outdoor activities in nature during the weekends may help foster nature connectedness.

### 5.4. Limitations

Our study had several limitations. First, we used self-reported data. Potential reporting bias can not be ruled out. Some people may have been afraid to disclose their poor mental health conditions, so wellbeing measures may have been inflated. In addition, we introduced the research topic when recruiting participants. It is possible that people who cared about urban green spaces responded to the survey. They may have had higher nature connectedness and nature contact than the general public. For these reasons, the effects of the observed pathways may have been overestimated. Our data may not have represented the general public due to its online administration. Future studies should use stratified sampling and offline investigation approaches to help reduce the risk of bias.

We also could not obtain real-time data for childhood variables. Thus, longitudinal studies are warranted. For example, starting an investigation among elementary school students that measures their current residential green space and nature contact and subsequently records their nature connectedness and nature contact every few years would help re-examine the observed pathways more precisely. Additionally, future studies may consider objective historical data, such as using Normalized Difference Vegetation Index (NDVI) values to measure residential green space. Further, some schools have started testing and recording students’ mental health conditions at different ages. These data may be available under a legal procedure to measure changes in mental health over time. Some of our variables were measured with single items, such as childhood residential green space and nature contact. This is because there is no widely accepted questionnaire or method for such variables. Although we tried to follow the measurements in previous research, the reliability of these measurements remains unclear. Future studies may need to address this issue by exploring appropriate dimensions or developing new questionnaires.

Our model was examined using data captured in a cross-sectional investigation. The causal relationships between variables were based on theoretical assumptions drawn from previous literature. Controlled trials are needed to examine the hypothesized causal relationships. In addition, though distinctive pathways were observed in the model, the effects of some indirect pathways were weak. Our model explained a small percentage of the total variation in adult nature contact and adult mental wellbeing. Additional mechanisms need to be investigated. Additional variables that might help to extend the framework include environmental disturbances, social cohesion, and physical activity, which may link residential green spaces with mental wellbeing [11,81,82].

People of different ages from 18 to 68 were involved in the current study. Given the rapid urbanization process in China, the characteristics of childhood residential green spaces may vary by generation. Older generations’ childhood residential green spaces may be wilder, while those of younger participants may be more cultivated or manicured. Thus, the impact of childhood residential green spaces may reflect qualitatively different experiences [83]. In addition, memory may have been a problem. Younger participants could have recalled their childhoods more easily than older participants. Although we conducted a multigroup analysis to test for differences by age, such biases may not have been eliminated completely. Future research should select suitable subjects to control the time intervals between past variables and current reports. Recruiting people of similar ages may be one solution.

## 6. Conclusions

The current study examined the role of childhood residential green space on adult mental health via multiple mediators. Adult nature contact and nature connectedness played important roles in linking childhood residential green space and adult mental wellbeing. Since childhood residential space can be an important source for children’s nature contact, policymakers are advised to improve residential green space as well as green recreational facilities for children to realize the potential of urban green spaces in health and health-promoting behavior.

## Figures and Tables

**Figure 2 behavsci-12-00084-f002:**
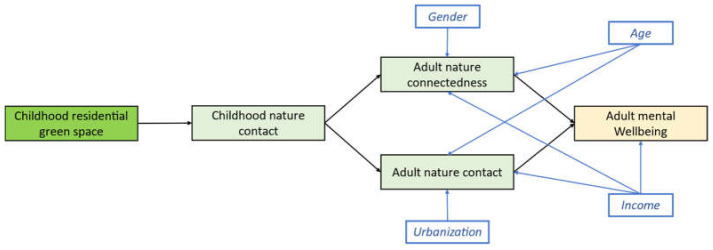
Initial model tested, including control variables.

**Figure 3 behavsci-12-00084-f003:**
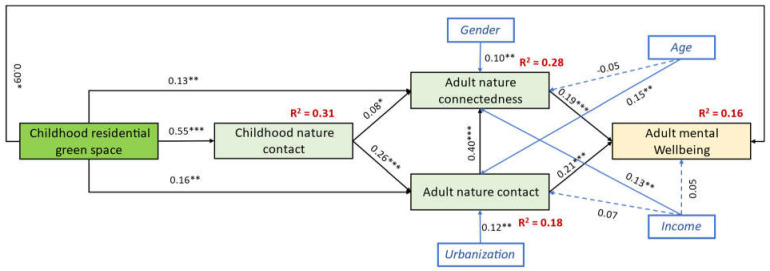
Final SEM with standardized coefficients (β). Solid lines indicate statistically significant pathways; dotted lines indicate statistically non-significant pathways. *, *p* < 0.05; **, *p* < 0.01; ***, *p* < 0.001. R^2^, explained variance.

**Table 1 behavsci-12-00084-t001:** Participant characteristics (*n* = 752).

Variable	Category	Mean (SD)	Range (Min–Max)	Percentage (N)
Urbanization rate	-	64% (0.09)	50.05–100%	-
Age	-	31.15 (10.45)	18–68	-
Gender	Male	-	-	56.38% (424)
	Female	-	-	43.62% (328)
Income	None	-	-	19.41% (146)
	0–3000	-	-	6.38% (48)
	3001–5000	-	-	18.35% (138)
	5001–7000	-	-	15.30% (115)
	7001–9000	-	-	13.56% (102)
	9001–11,000	-	-	12.77% (96)
	11,001–13,000	-	-	4.65% (35)
	13,001–15,000	-	-	3.72% (28)
	More than 15,000	-	-	5.85% (44)
Childhood residential green space	Extremely low	-	-	6.10% (47)
	Low	-	-	20.61% (155)
	Moderate	-	-	21.41% (161)
	High	-	-	25.93% (195)
	Extremely high	-	-	25.80% (194)
Nature connectedness	-	27.23 (5.58)	7–35	-
Mental wellbeing	-	17.94 (5.61)	0–25	-
Childhood nature contact	Never	-	-	3.19% (24)
	A few days	-	-	17.42% (131)
	Half of days	-	-	25.13% (189)
	Most of days	-	-	25.39% (191)
	Almost every day	-	-	28.86% (217)
Adult nature contact	Never	-	-	1.46% (11)
	A few days	-	-	26.33% (198)
	Half of days	-	-	32.18% (242)
	Most of days	-	-	29.92% (225)
	Almost Every day	-	-	10.11% (76)

**Table 2 behavsci-12-00084-t002:** Correlations (Spearman and point-biserial) between the measurements.

Variables	1	2	3	4	5	6	7	8	9
1. Gender	1								
2. Age	**0.16 *****	1.00							
3. Income	−0.03	**0.52 *****	1.00						
4. Urbanization rate	−0.01	0.01	**0.10 *****	1.00					
5. Childhood green space	−0.05	−0.03	**0.12 *****	−0.04	1.00				
6. Child nature contact	−0.01	**0.10 *****	**0.17 *****	−0.06	**0.58 *****	1.00			
7. Adult nature contact	−0.03	**0.22 *****	**0.24 *****	**0.08 ****	**0.30 *****	**0.36 *****	1.00		
8. Adult nature connectedness	**0.06 ***	**0.15 *****	**0.23 *****	0.04	**0.32 *****	**0.32 *****	**0.48 *****	1.00	
9. Adult mental wellbeing	−0.06	0.07	**0.16 *****	0.06	**0.24 *****	**0.25 *****	**0.35 *****	**0.39 *****	1.00

Numbers in cells indicate Spearman’s rho. *, *p* < 0.1; **, *p* < 0.05; ***, *p* < 0.01. Significant correlations show in bold, *p* < 0.1.

**Table 3 behavsci-12-00084-t003:** Standardized total and indirect effects of childhood residential green space on core variables.

Variables	Total Effect	*p*	Indirect Effect (95%CI)	*p*
Childhood nature contact	0.55	<0.001	-	-
Adult nature contact	0.30	<0.001	0.15	<0.001
Adult nature connectedness	0.30	<0.001	0.16	<0.001
Adult mental wellbeing	0.22	<0.001	0.12	<0.001

**Table 4 behavsci-12-00084-t004:** Indirect pathways between childhood residential green space and adult mental wellbeing.

Pathways	Effect	*p*
Childhood green space → adult nature connectedness → adult mental wellbeing	0.025	<0.001
Childhood green space → adult nature contact → adult mental wellbeing	0.033	<0.001
Childhood green space → childhood nature contact → adult nature connectedness → mental wellbeing	0.008	0.033
Childhood green space → childhood nature contact → adult nature contact → mental wellbeing	0.031	<0.001
Childhood green space → adult nature contact → adult nature connectedness → mental wellbeing	0.012	<0.001
Childhood green space → childhood nature contact → adult nature contact → adult nature → adult nature connectedness → mental wellbeing	0.011	<0.001

## Data Availability

The data is available upon request from the corresponding author.

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
