# Peer review of "From Childhood Residential Green space to Adult Mental Wellbeing: A Pathway Analysis among Chinese Adults"

_behavsci, 2022, doi:10.3390/bs12030084_

Round 1
Reviewer 1 Report
Dear authors,
I found the topic of your article very interesting, and I enjoy reading it. I have some comments and reflections to help improve your research.
I hope to contribute positively to your work.
Introduction:
- I liked the organization in topics, it helps the reader to follow the logic of your argument.
- Did you follow a specific theoretical framework, or was your research just based on a scattered body of prior knowledge?
- I think Figures 1 and 2 are not necessary; You explain the expected paths well and you can just refer to Figure 3.
- The order in which the first and second indirect paths are presented in the text is not the same as in the figure. I think it could be easier to read if you correct the order.
- “From childhood contact with nature to adult mental well-being” – Do you consider only indirect paths? Just out of curiosity: Isn't it possible that childhood contacts with nature directly influence adult mental well-being? Could it be interesting, in addition to the total mediation model that you hypothesize, to test the partial mediation model and the non-mediation model?
- lines 98-99 – I would like to know more about what hypotheses you make about the possible influence of covariates in your model. For example, how can gender influence the relationship between adult connection to nature and adult mental well-being?
Materials and methods:
- Study design - it may be important to mention that the study is cross-sectional
- A curiosity: regarding the inclusion criteria, did you control the generation of participants? Are the participants in the present study from the previous generation that had more contact with nature (as you mentioned in the introduction) or are they from the current generation that lives in the era of urbanization? Or both? Could this aspect influence the data? In what way? In addition, have you controlled for the time interval between childhood and the moment when participants answer the questionnaire? Could this lead to different recall experiences (i.e., 18-year-old participants may have better memory of their contact with nature when they were 5-10 years old than 40-year-old participants)?
- What does extremely low or extremely high residential green space mean? Having access to a garden at home, having a garden 2 meters from the house? Maybe you can explain better.
- I am concerned about the quality of some variables. Measuring a variable based on just one item can be a very poor approach. A scale must have at least three items. How do you guarantee that the questions used can capture the variable you want well?
- What are the urbanization rate categorizations? For example, what does an urbanization rate of 0.60 mean?
- Congratulations on the robustness of the chosen data analysis method.
- In structural equation models, there are observed variables and latent variables. It was not clear to me what are the observed variables and the latent variables of your model, can you explain better?
- I would like to see, in the introduction, literature that could support the changes you made to the model (e.g., new paths and hypotheses) after you found that the fit was weak in the initial model.
Results:
- The results are described with scientific rigor.
- Could you put the ranges of the variables in the legend of table 1?
- In the supplementary files, it is difficult to see the differences between the figures. Could you draw arrows with bigger tips?
Discussion:
- I think that the results were well reflected in the discussion and well contrasted with the literature.
- Data were collected during a pandemic period. Could this have affected the results? In what way? For example, has the pandemic affected adults' contact with nature?
20.Regarding implications and suggestions for practice, I would like to see suggestions and guidelines for parents and schools (e.g., what curriculum programs could be developed).
- I would like to see more details regarding the solutions to the limitations. For example, how would you run the longitudinal study? Would you collect data from participants from childhood to adulthood? Over a period of 15 years for example?
- Future directions: What other variables do you suggest adding to the explicative model?
Apart from these comments, I consider your article valuable for the research community and I propose to be published after address these issues.
Reviewer 2 Report
This is an interesting paper about an important topic. The authors did a nice job on this. Here are some comments that I feel could help strengthen the paper.
Abstract: Please include the year the data were collected.
Intro:
1) The last sentence in the first paragraph needs to be revised. It seems like there needs to be an "and" before the "are" after the comma.
Methods:
1) 2.3.3 should start with "following" and not "followed."
2) Did the authors translate all of these measures or did they use already accepted translations?
3) Was the urbanization rate the urbanization where they currently live or where they lived when they grew up?
Results:
1) Can you interpret the estimates specifically? It would help to show how much of an effect each aspect had.
Discussion:
1) The limitations section is brief. You mention reporting bias, but I think you need to specifically address recall bias given the amount of time that has passed between the exposure and outcome. Please also acknowledge in your limitations the lack of representation of the sample given that it is a convenience sample. Based on who is more likely to respond to your survey (people who may care about nature?), how might this likely bias your estimates toward or away from the null?
Round 2
Reviewer 1 Report
Dear authors,
I am very happy with the significant improvement of your paper and glad that you considered my suggestions. I only have a few small comments and reflections left.
16.Could you put the ranges of the variables in the legend of table 1?
Response: There are only four variables that are not in category, we have added their ranges in the table now.
New comment: I think you can add a "-" in the blanks. It will make the table easier to read.
- I would like to see more details regarding the solutions to the limitations. For example, how would you run the longitudinal study? Would you collect data from participants from childhood to adulthood? Over a period of 15 years for example?
Response: Tracking research will be difficult. The data in childhood can be obtained from some platforms. We have proposed suggestions in the limitation (lines: 410-415)
They read as: “We could not obtain the real-time data for the past variables. Thus, the investigation for childhood can be inaccurate. Future studies may consider historical data, such as using the historic Normalized Difference Vegetation Index (NDVI) to measure residential greenspace. Besides, some platforms may also be useful. For example, some schools
have started testing and recording students’ mental health conditions at different ages. These data may be available under a legal procedure.”
New comment: Ok, perfect. However, why would a longitudinal study be difficult? You could start a new study today, with children, and follow those children over a period of X years.
Congratulations on your paper! Thank you, obrigada.
